# Molecular Detection of *Trypanosoma* spp. in Questing and Feeding Ticks (Ixodidae) Collected from an Endemic Region of South-West Australia

**DOI:** 10.3390/pathogens10081037

**Published:** 2021-08-16

**Authors:** Anna-Sheree Krige, R. C. Andrew Thompson, Anke Seidlitz, Sarah Keatley, Julia Wayne, Peta L. Clode

**Affiliations:** 1UWA School of Biological Sciences, The University of Western Australia, 35 Stirling Highway, Crawley, WA 6009, Australia; peta.clode@uwa.edu.au; 2School of Veterinary and Life Sciences, Murdoch University, 90 South Street, Murdoch, WA 6150, Australia; a.thompson@murdoch.edu.au (R.C.A.T.); s.keatley@murdoch.edu.au (S.K.); 3School of Environmental and Conservation Sciences, Murdoch University, 90 South Street, Murdoch, WA 6150, Australia; anke.seidlitz@gmx.net; 4Department of Biodiversity, Conservation and Attractions (DBCA), Locked Bag 2, Manjimup, WA 6258, Australia; julia.wayne@dbca.wa.gov.au; 5Centre for Microscopy, Characterisation and Analysis, The University of Western Australia, 35 Stirling Highway, Crawley, WA 6009, Australia

**Keywords:** trypanosomes, PCR, prevalence, infection, ticks, vectors

## Abstract

A growing number of indigenous trypanosomes have been reported to naturally infect a variety of Australian wildlife with some species of *Trypanosoma* implicated in the population decline of critically endangered marsupials. However, the mode of transmission of Australian trypanosomes is unknown since their vectors remain unidentified. Here we aimed to fill this current knowledge gap about the occurrence and identity of indigenous trypanosomes in Australian invertebrates by conducting molecular screening for the presence of *Trypanosoma* spp. in native ticks collected from south-west Australia. A total of 231 ticks (148 collected from vegetation and 83 retrieved directly from 76 marsupial hosts) were screened for *Trypanosoma* using a High-Resolution Melt (HRM) qPCR assay. An overall *Trypanosoma* qPCR positivity of 37% (46/125) and 34% (26/76) was detected in questing ticks and host-collected (i.e., feeding) ticks, respectively. Of these, sequencing revealed 28% (35/125) of questing and 28% (21/76) of feeding ticks were infected with one or more of the five species of trypanosome previously reported in this region (*T. copemani, T. noyesi, T. vegrandis, T. gilletti, Trypanosoma* sp. ANU2). This work has confirmed that Australian ticks are capable of harbouring several species of indigenous trypanosome and likely serve as their vectors.

## 1. Introduction

Trypanosomes are universal, unicellular blood-borne protozoan parasites of the class Kinetoplastea, with some species of veterinary and medical significance due to their ability to cause debilitating infections in a broad range of hosts. With a digenetic life cycle, transmission between mammalian hosts is typically dependent on haematophagous invertebrates that function as vectors [1]. In countries other than Australia, the common vectors for mammalian trypanosomes include haematophagous representatives within the order Diptera: Glossinidae (tsetse flies), Tabanidae (tabanids) and Ceratopogonidae (biting midges). Other vectors include Siphonaptera (fleas) and members of the heteropteran family, Reduviidae (triatomines) [1,2,3,4,5].

Within Australia, a growing number of indigenous trypanosomes have been found to naturally infect a variety of native wildlife [6,7,8,9,10,11]. Of particular concern is the reported association of species of indigenous trypanosomes as the causative agents in the deteriorating health, and subsequent population decline, of certain wildlife such as the woylie (syn. brush-tailed bettong, *Bettongia penicillata*) [9,12,13]. A native marsupial whose geographical distribution has become restricted to a region of south-western Australia, the woylie has undergone a greater than 90% decline in population numbers since 1999, affirming it as critically endangered on the IUCN list of threatened species [13]. Whilst there have been various hypothetical proposals for this dramatic decline, their present spatial distribution suggests the potential role of an infectious agent [13,14]. Recent surveys have confirmed the woylie as capable of harbouring at least five indigenous trypanosome species: *Trypanosoma copemani, T. gilletti, T. noyesi* and *T. vegrandis*, as well as a currently undescribed species, tentatively designated as *Trypanosoma* sp. ANU2 [9,15]. One genotype of *T. copemani* has been detected in woylie tissues and subsequently associated with tissue pathology [12]. 

Ticks (Ixodidae) are arthropods with a life history that involves four developmental stages, three of which require a blood meal from a vertebrate host [16]. It is during this act of blood-feeding that ticks are capable of acquiring several infectious agents from a reservoir host including bacteria, viruses and parasites. Moreover, some of these pathogens have the potential to persist within the ticks by trans-stadial survival (i.e., between different developmental stages), making ticks globally recognised as important vectors for an array of infectious diseases in humans and animals [17]. Ticks have persisted for several years as the prime suspect responsible for transmitting Australian trypanosomes between wildlife. Unfortunately, since previous studies have focused solely on the opportunistic screening of blood-fed ticks collected directly from infected vertebrate hosts, the lack of conclusive evidence has delayed any advancement in our understanding of the life histories of Australia’s trypanosomes [18]. Consequently, the vectors responsible for maintaining Australia’s *Trypanosoma* spp. life cycles and the ubiquitous persistence of several species within native fauna remains unknown [19].

A recent study investigating the presence of *T. noyesi*, an Australian parasite of special biosecurity interest due to its genetic proximity to the South American human pathogen *T. cruzi* [20,21], revealed ticks collected from the vegetation (i.e., questing) as capable of harbouring this parasite despite the apparent absence of a blood meal [22]. Furthermore, this study determined that several different morphotypes for *T. noyesi* were present within the combined smeared gut and salivary gland contents of questing ticks, including the trypomastigote form; the developmental stage infectious to vertebrate hosts. 

With an aim to fill the current gaps in knowledge concerning the identity and prevalence of trypanosomes within Australian ticks, the focus of the present study was to molecularly screen questing and feeding ticks, collected from an endemic region of south-west Australia, for indigenous species of *Trypanosoma* parasites. All questing ticks were examined in a previous study [22] to detect and visualise *T. noyesi.* The aim of the present study was to survey Australian ticks for additional species of *Trypanosoma* common to the native wildlife that inhabit this region of south-western Australia.

## 2. Materials and Methods

### 2.1. Study Sites, Sampling and Tick Identification

Sampling was conducted within the Upper Warren Region (UWR) of south-western Australia (approximately 144,000 ha of nature reserve and forested areas). Questing ticks were collected by opportunistic sampling—ticks were removed from clothing and equipment on which they had crawled. Given the observable size of nymphs and adults, these life stages were further collected from the ground as and when observed. Collection occurred in 34 different forest block sites from 15 geographical locations within this region. Sampling was in concurrence with wildlife surveys conducted by the Department of Biodiversity, Conservation and Attractions (DBCA) (formally Department of Parks and Wildlife) Fauna Taking (Scientific or Other Purposes) Licence (08-000995-3; FO25000048; FO25000173) in collaboration with Murdoch University, a joint collaboration throughout the spring and summer months of September-December 2018. During these fauna monitoring surveys, several species of wildlife were trapped under DBCA permit number DBCA AEC 2018_22F during which time feeding ticks were removed. Wildlife were trapped within five geographical localities within the UWR. Upon collection, all ticks were immediately preserved in 70% ethanol and stored at ambient temperature during field work before subsequently stored at 4 °C in the laboratory until downstream analyses. Ticks were identified morphologically in accordance with published tick identification keys [23,24]. 

### 2.2. DNA Extraction and HRM-qPCR Detection of Trypanosoma spp.

Following identification, the external surface of each tick was cleaned in a series of 10% sodium hypochlorite, followed by 70% ethanol and a final rinse in sterile phosphate buffered saline (PBS; 10 mM Na_2_HPO_4_, 150 mM NaCl at pH 7.4) prior to DNA extraction. Ticks were bisected as previously described [22]. The total genomic DNA was extracted from individual bisected ticks using a DNeasy Blood & Tissue kit (QIAGEN, Hilden, Germany) following the manufacturers recommendations (QIAGEN Supplementary Protocol: Purification of total DNA from insects): overnight digestion at 56 °C; elution buffer AE decreased to between 30–40 μL (depending on the size of the tick); double elution step to increase DNA yield. Each DNA extraction batch contained blank controls subsequently checked by PCR for contamination. *Trypanosoma* spp. detection was performed on genomic DNA extracts from individual adult and nymph questing ticks (*n* = 123) and two questing larvae pools (*n* = 15 and 10). For feeding ticks, individual adults, nymphs and larvae (*n* = 72) and four feeding larvae pools (*n* = 2, 3, 2 and 4) were analysed. Screening was employed for a 250 bp region using a High-Resolution Melt (HRM) real-time qPCR assay that targets the 18S rRNA gene for trypanosomatids, with amplification parameters followed in accordance with the cycling protocol [25]. All HRM-qPCRs were run with positive controls obtained from cloned DNA from three *Trypanosoma* species: *T. copemani, T. noyesi* and *T. vegrandis*, which are species endemic to the region in which sampling took place. Negative controls included extraction blanks, PCR-grade H_2_O, no template control (NTC) (i.e., reagents only) and *Leishmania macropodum* [22]. All analyses were carried out in duplicate. Post amplification melt curve analysis allowed for initial differentiation of *Trypanosoma* species [25] that was subsequently confirmed with sequencing. 

### 2.3. Confirmation of qPCR Positive Results by DNA Sequencing

PCR products were visualised on 1.5% agarose gels stained with SYBR Safe (Invitrogen, California, USA). Amplicons were purified using Agencourt AMPure Purification System (Beckman Coulter, California, USA). Sequencing was carried out at the Western Australian State Agricultural Biotechnology Centre (SABC) at Murdoch University, using an ABI Prism™ BigDye v3.1 Cycle Sequencing Kit (Applied Biosystems, California, USA) and an ABI 3730 96 capillary machine. Forward and reverse DNA strands were sequenced and subsequently analysed in Geneious v10.2.2 (https://www.geneious.com). Sequence identification was performed using BLAST (http://blast.ncbi.nlm.nih.gov/Blast.cgi) against the NCBI nucleotide (nt) database. Phylogenetic analyses of the amplified section of 18S rRNA was conducted for *Trypanosoma* spp. (*T. copemani, T. noyesi, T. vegrandis/T. gilletti* and *Trypanosoma* sp. ANU2) and *Bodo* DNA detected within ticks. Maximum likelihood (ML) trees were generated using the HKY85 genetic distance model with bootstrap resampling using 1000 replicates and constructed using PHyML v3.2 in Geneious [26,27]. 

### 2.4. Statistical Analysis

Statistical analyses were performed using a chi-squared test (Minitab 18 Statistical Software) to examine the difference in *Trypanosoma* spp. infection between questing and feeding ticks. Due to the small sample size, a Fisher exact test was used to compare the infection rate between life stages: larvae, nymphs and adults collected from vegetation against those retrieved from marsupial hosts. Significance level (alpha) was set to 0.05.

## 3. Results

### 3.1. Tick Collection and Identification

A total of 231 ticks were collected in the UWR and molecularly screened for Australian species of *Trypanosoma* using a trypanosomatid-specific HRM-qPCR assay. Of these, 148 questing ticks (*n* = 25 larvae, 82 nymphs, 21 adult males and 20 adult females) were retrieved from vegetation. Questing adult and nymph ticks were identified as *Amblyomma triguttatum*, *Ixodes australiensis* and *I. myrmecobii*, with larvae identified as *Ixodes* spp. Feeding ticks (*n* = 83; 21 larvae, 21 nymphs, 1 adult male and 40 adult females) were collected from a total of 76 marsupial hosts including the woylie (*n* = 59), brush-tailed possum *Trichosurus vulpecula* (*n* = 15), quenda *Isoodon obesulus* (*n* = 1) and tammar wallaby *Macropus eugenii* (*n* = 1). Adult and nymph feeding ticks were predominantly identified as *I. australiensis, I. myrmecobii, I. tasmani* and the newly described *I. woyliei* [28], with a single *A. triguttatum* and *I. fecialis* retrieved from a woylie and quenda, respectively. Larval ticks were identified as *Amblyomma* spp. and *Ixodes* spp. The sampling of both questing and feeding ticks were opportunistic and therefore not standardised with respect to collection area or time. The developmental stages of collected tick species are represented in Table 1. Notably woylies carried the highest diversity of tick species detected (Table 1). *Ixodes australiensis* was the most common tick species retrieved from the woylie, whereas *I. tasmani* was the prevalent species collected from the brush-tailed possum.

### 3.2. HRM-qPCR Detection of Trypanosoma spp. in Questing and Feeding Ticks

The recently published HRM-qPCR assay specific for trypanosomatids [25], revealed 35% (71/201) of collected ticks as positive for *Trypanosoma* DNA. Data was confirmed by qPCR signal within the amplification parameters for *Trypanosoma* spp., as previously established [25], and the presence of an amplicon of ~250 bp. The difference between ticks retrieved from vegetation (37%; 46/125) compared to feeding ticks removed from hosts (33%; 25/76) was insignificant. The distribution of *Trypanosoma* qPCR data across the different life stages for questing and feeding ticks is presented in Table 1. In the feeding ticks collected from marsupial hosts, the detection of *Trypanosoma* DNA was highest in the brush-tailed possum (53%; 8/15) followed by the woylie (22%; 13/59). Neither the quenda or tammar wallaby sampled in this study were qPCR positive; however, only a single marsupial of each species was opportunistically sampled.

### 3.3. Sequence Based Identification of Trypanosoma spp. in Ticks

Sanger sequencing established that 28% of questing (*n* = 35; 9 adults and 26 nymphs) and 28% of feeding ticks (*n* = 21; 7 adults, 10 nymphs and 4 larval pools) were infected with one or more of the trypanosome species previously documented in the region [9,15]. These species include *T. copemani, T. noyesi, T. vegrandis*/*T. gilletti* species complex [15], and *Trypanosoma* sp. ANU2. Table 2 displays the occurrence of these *Trypanosoma* spp. in the questing and feeding ticks sampled in this study. Representatives for each species of trypanosome detected in this study are presented in Figure 1. Sequence identity from BLAST results identified *T. copemani* in 11% (14/125) and 8% (6/76) of questing and feeding ticks, respectively. The *Trypanosoma* 18S rRNA sequence amplified from ticks was found to share 95.24–100% sequence similarity to partial sequences from *T. copemani* genotypes. *T. noyesi* was confirmed in 5% (6/125) of questing ticks as previously published [22] and 1% (1/76) of feeding ticks. The region amplified shared 97.97–99.19% sequence similarity to deposited GenBank sequences for *T. noyesi* and *T. vegrandis/T. gilletti* and was identified in 10% (13/125) and 15% (11/76) of questing and feeding ticks, respectively. Sequences shared 97.73–100% similarity to deposited sequences corresponding to species within this complex. Analysis concluded that 2% (2/125) questing and 4% (3/76) feeding ticks contained DNA for *Trypanosoma* sp. ANU2 (98.46–100% similarity). One questing tick generated sequences for *T. copemani* and *T. vegrandis* DNA indicating a mixed infection. Whilst data is insufficient to deduce meaningful phylogenetic inferences between *Trypanosoma* spp. and different tick species (Figure 1), sequence disparities were apparent and primarily the result of small sequence variants including SNPs. *T. noyesi* and *T. vegrandis/T. gilletti* sequences with reduced sequence similarity shared SNP’s at the terminal ends. Although *T. noyesi*-positive ticks were of the species *A. triguttatum*, a single questing tick contained a nucleotide insertion that was not shared by the remaining *T. noyesi*-positive ticks. Similarly, the sequence from a questing *I. woyliei* contained one nucleotide insertion that was not shared by other *T. copemani*-positive ticks. SNP and nucleotide insertions may account for different trypanosome genotypes, however no significant nucleotide diversity was correlated with different tick species. The *Bodo* DNA detected in *A. triguttatum*, *I. australiensis*, and *I. myrmecobii* ticks shared similarity with *Bodo* spp. (Appendix A). All questing ticks were previously examined for the presence of *T. noyesi* DNA [22] with amplified sequences from the study available in GenBank. New sequences obtained in the present study have been deposited in GenBank for *Trypanosoma* spp. (accession numbers: MW881302-MW881349) and *Bodo* spp. (accession numbers: MZ669864-MZ669882) (Appendix A).

Finally, among the feeding ticks screened, *Trypanosoma* spp. DNA was frequently detected in *I. tasmani* (50%; 5/10). In comparison, the dominant detection of *Trypanosoma* spp. DNA in questing ticks was within *A. triguttatum* (89%; 31/35). Overall, the tick developmental phase in which *Trypanosoma* DNA was most prevalent in both feeding and questing ticks was the nymphal life stage.

### 3.4. Statistical Analyses

The difference between the percentage of *Trypanosoma* infected questing ticks (28%) and *Trypanosoma* infected feeding ticks (~28%) was not statistically significant (*p* = 0.5408). Similarly, the difference between infected nymphs (questing vs. feeding) and adult ticks (questing vs. feeding) was not statistically significant. However, the difference between the percentage of infected questing larvae (0%) and infected feeding larvae (38%) was statistically significant (*p* = 0.0008).

## 4. Discussion

This study provides molecular evidence for the presence of five Australian *Trypanosoma* spp. (*T. copemani, T. noyesi, T. vegrandis, T. gilletti* and *Trypanosoma* sp. ANU2) from questing and feeding ticks (genera *Amblyomma* and *Ixodes*) collected from south-west Australia. These data provide the first survey of several *Trypanosoma* spp. in Australian questing ticks, as well as a direct comparison with that of feeding ticks collected at the same time and location.

Trypanosomes of Australian wildlife are significantly understudied despite several reports suggesting that some species might be pathogenic, affecting the fitness and consequently the population numbers for certain native hosts [12]. A major knowledge gap persists in regard to understanding the Australian *Trypanosoma* life cycle(s) since the vectors have not been identified. Whilst ticks have been proposed as vectors due to the frequency in which they are found on wildlife, in addition to sporadic reports of flagellate forms within opportunistically collected fed ticks, to date there has been a scarcity of published reports detailing the screening of any potential invertebrate vector candidates for Australian trypanosomes [18,19,20,22,29,30]. This is despite the continuing discovery of new species and the revelation that many indigenous species of *Trypanosoma* appear to be ubiquitously distributed across the Australian continent and found in a remarkably diverse range of vertebrate hosts [6,11,15,31,32].

The presence of *Trypanosoma* DNA in questing ticks suggests that ticks may serve as potential vectors for indigenous trypanosomes in Australia, hence contributing to the extensive distribution of these *Trypanosoma* spp.; corresponding with the common and widespread dissemination of ticks throughout the continent [23,24]. *Trypanosoma* DNA was detected in all examined stages (larvae, nymph, adult) of both feeding and questing ticks, however statistical comparison between stages is difficult because of the differences in ratios among the collected developmental stages of ticks. Hence, a direct comparison of the unequally distributed stages of ticks in these two groups was deemed impractical. Nevertheless, our observations are in agreement with the known biology of ticks: adults are the life stage more frequently found on hosts, whilst immature nymphs are more prevalent on vegetation. Interestingly, a number of larvae were also retrieved from two host species in this study. In the Australian ecosystem, these hosts, woylie and brush-tailed possum, are considerably smaller when compared to the larger macropods that inhabit the environment. As inferred for the ecology of ticks, these smaller marsupials are opportunistic hosts for immature stage ticks such as larvae, due to their close proximity to the ground (i.e., leaf litter) where immature stages are generally found [16].

The overestimation for trypanosomatid DNA by HRM-qPCR (37%; Table 1) compared to the *Trypanosoma* DNA confirmed by subsequent sequencing (28%) was accounted for as a result of the presence of *Bodo* DNA, as previously reported [22]. Whilst further evaluation concerning the occurrence of *Bodo* DNA within ticks was beyond the scope of this study, the presence of this kinetoplastid may be attributed to various factors, including contaminated moist leaf litter/soil and the consumption of water—environments in which these free-living protozoa are often located. Statistical analyses suggested significance in regard to the frequency rate of *Trypanosoma* DNA in feeding larvae when compared to the absence of infection in questing larvae. This difference may be explained in terms of tick biology, where questing larvae are yet to receive their first blood meal and hence have not had an opportunity to acquire a *Trypanosoma* infection. The presence of *Trypanosoma* DNA in feeding larvae suggests acquisition from a parasitaemic host. The occurrence of *Trypanosoma* spp. infection in wildlife collected from this region has been previously detected at high prevalence (>80%) [12]. Consequently, when we consider a blood meal acquired infection, the presence of *Trypanosoma* DNA in feeding ticks directly removed from infected hosts within this region is not unusual [18]. However, when the occurrence in feeding larvae is subsequently compared to the high detection of *Trypanosoma* DNA in questing nymphs, these data may suggest that the larvae, having previously fed on a parasitaemic host, is capable of maintaining the acquired infection between developmental stages and therefore supporting the hypothesis of trans-stadial transmission. Hence, we can assume that the infection acquired during the larval stage has every possibility of persisting from larval to nymphal stage. This may explain the significant number of questing nymphs found infected with *Trypanosoma* DNA in this investigation.

Infection with more than one trypanosome was confirmed for a single questing tick, however the potential for mixed infections in other ticks cannot be excluded. HRM-qPCR analysis provided one indication of mixed *Trypanosoma* spp. DNA. Co-infection with *T. copemani* and *T. vegrandis* or *T. gilletti* (undistinguishable from each other by the amplified fragment) was detected in a questing immature nymphal stage. This suggests that the infection was likely acquired from feeding at the larval stage on a host harbouring a blood infection comprising of these *Trypanosoma* species. This theory is plausible given ongoing reports of multiple infections for several species of trypanosome within Australian wildlife, such as the woylie [9], a marsupial endemic to the area sampled.

In this study, the *Trypanosoma* species identified within feeding ticks revealed *T. vegrandis/T. gilletti* as most prevalent (15%), followed by *T. copemani* (8%), *Trypanosoma* sp. ANU2 (4%) and *T. noyesi* (1%). This prevalence gradient directly corresponds with the statistical data from a recent study investigating trypanosome incidence in woylies from the same area sampled [9]. Interestingly, of the 21 (28%) feeding ticks that contained *Trypanosoma* DNA, only one was DNA positive for *T. noyesi* (Table 2), compared to the six questing ticks positive for *T. noyesi* [22]. Whilst not statistically significant, it can be expected that the percentage of infection would be higher in feeding ticks due to the increased likelihood of a blood meal acquired infection from a host exhibiting parasitaemia. However, since the infection status of the hosts at the time in which feeding ticks were collected was unknown, we cannot confirm whether, corresponding with a potential absence of *T. noyesi* infected hosts, these feeding ticks were simply free from *T. noyesi* DNA. Incidentally, woylies, the primary host species surveyed for feeding ticks in this study, have been reported to harbour a low infection rate for *T. noyesi*, particularly when compared with the other species of trypanosome for which the marsupial is a known carrier [9,12,20]. Consequently, the outstanding question remains as to the identity of the main reservoir host(s) for *T. noyesi* within the Australian ecosystem. Of further interest is the single feeding tick positive for *T. noyesi* DNA which belonged to the species *A. triguttatum*; the same species of tick reported as positive for *T. noyesi* in the recently published study concerning questing ticks collected from the same area [22].

In conclusion, this study adds to the growing evidence that ticks may serve as vectors for Australian trypanosomes. A limitation of this study is the sample size, whereby sampling was neither systematic nor exhaustive; the sample size for certain tick species was small and given the variety of prospective wildlife hosts in this area compared to the number of species opportunistically sampled during targeted surveys (i.e., woylie-specific), confirms that overall the hosts of feeding ticks collected in this study were understudied. A comprehensive representation for the prevalence of *Trypanosoma* in feeding ticks compared to questing ticks remains to be elucidated, with the results generated from this study providing a snapshot to guide a more thorough systematic, year-round, investigation.

## Figures and Tables

**Figure 1 pathogens-10-01037-f001:**
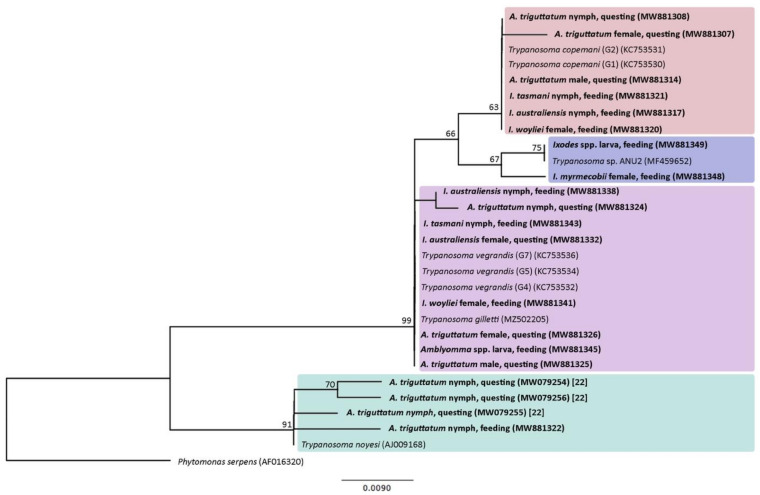
ML tree constructed from a 193 bp alignment of *Trypanosoma* 18S rRNA partial sequences. Numbers represent bootstrap support generated from 1000 replications. Support values above 60% are indicated. GenBank accession numbers in brackets. Sequences generated in the previous investigation [22] and this study are in bold type.

**Table 1 pathogens-10-01037-t001:** Summary of tick identification and the incidence of *Trypanosoma* positivity per developmental stage (larva, nymph, adult male, adult female) for ticks collected from vegetation (i.e., questing) and hosts (i.e., feeding).

Tick Species	Life Stage				Total	Host Species
	Larva	Nymph	Male	Female		
Questing						
*Amblyomma triguttatum*	0	78	10	9	97	n/a
*Ixodes australiensis*	0	4	11	4	19	n/a
*Ixodes myrmecobii*	0	0	0	7	7	n/a
*Ixodes* spp.	25	0	0	0	25	n/a
	*n*/qPCR+/%					
Total	25/0/0.0 *	82/30/36.6	21/8/38.1	20/8/40.0	148/46/31.1	
DNA extracts					125/46/36.8	
Feeding						
*Amblyomma triguttatum*	0	1	0	0	1	Woylie
*Amblyomma* spp.	20	0	0	0	20	Tammar wallaby, Woylie
*Ixodes australiensis*	0	7	1	18	26	Brush-tailed possum, Woylie
*Ixodes fecialis*	0	0	0	1	1	Quenda
*Ixodes myrmecobii*	0	2	0	9	11	Brush-tailed possum, Woylie
*Ixodes tasmani*	0	9	0	1	10	Brush-tailed possum
*Ixodes woyliei*	0	2	0	11	13	Woylie
*Ixodes* spp.	1	0	0	0	1	Woylie
	*n*/qPCR+/%					
Total	21/8/38.0 *	21/10/47.6	1/0/0.0	40/7/17.5	83/25/30.1	
DNA extracts					76/25/32.9	

qPCR, quantitative polymerase chain reaction. qPCR+, qPCR *Trypanosoma* positivity. * Larvae were pooled for DNA extracts.

**Table 2 pathogens-10-01037-t002:** Occurrence of *Trypanosoma* spp. in Australian ticks collected in this study.

	Questing Ticks	Feeding Ticks
*Trypanosoma copemani n* (%)	14 (11.2)	6 (7.9)
*Trypanosoma noyesi n* (%)	6 (4.8) [22]	1 (1.3)
*Trypanosoma vegrandis/T. gilletti n* (%)	13 (10.4)	11 (14.5)
*Trypanosoma* sp. ANU2 *n* (%)	2 (1.6)	3 (3.9)
Total *n* (%)	35/125 (28)	21/76 (27.6)

## Data Availability

The partial nucleotide sequences generated in this study have been deposited in the NCBI database and are publicly available under accession numbers MW881302-MW881349 and MZ669864-MZ669882 for *Trypanosoma* spp. and *Bodo* spp., respectively.

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
