# Peer review of "Molecular Detection of Trypanosoma spp. in Questing and Feeding Ticks (Ixodidae) Collected from an Endemic Region of South-West Australia"

_pathogens, 2021, doi:10.3390/pathogens10081037_

Round 1

Reviewer 1 Report

This study characterizes the distribution of five species of Trypanosoma parasites in questing ticks.  The data complement an earlier study that characterized the distribution of the same Trypanosoma species in feeding ticks.  Overall the results are incremental ,and some interesting issues are not developed.  For example, A) the concept of trypanosome presence in larval stages that have not taken a blood meal, and B) the presence of trypanosomatids other that the indicated five species – a point that is raised by the discovery elsewhere of bodonid DNA in ticks.  The latter observation is fascinating as bodonids are generally considered to be aquatic in nature with free-living and fish-pathogenic isolates.

Author Response

Response to Reviewer 1 Comments

The authors thank reviewer 1 for their positive comments concerning our study and acknowledge no suggested changes to the current manuscript.

Reviewer 2 Report

I reviewed the article titled: “Molecular detection of Trypanosoma spp. in questing and feeding ticks (Ixodidae) collected from an endemic region of south-west Australia” and I found it well-prepared. The authors presented the molecular detection of Trypanosoma spp. in questing and feeding ticks and for the first time provided a direct comparison between questing and feeding ticks collected from the same endemic region. Although the unsystematic form of sampling and the limited sample size are the main flaws of this research I consider this work as suitable to be published in Pathogens after minor revision. Please review the following points:

Materials and Methods:

[171] Please indicate in which statistical program the statistical analyses were performed.

Results:

[189] There's no need to repeat the information about permit number as it was mentioned in Material and Methods section.

[201] The Table 1 can be divided in two separate tables - one for questing ticks, the other one for feeding ticks.

[202] The abbreviations L, N, A are not necessary - these words are not too long to not be placed in the table. I suppose it will be easier to read it.

[203] In my opinion the adults column should be above male and female columns as the division between sexes is made only in this stage of ticks.

[203] Do the authors have an information about the presence of questing ticks on specific sites? I.e. the larvae was collected from the leaves, whereas adult forms from clothes? In my opinion the information about the exact localization of questing ticks at specific life stage should be indicated like the host species in case of feeding ticks.

[204] The information L n/qPCR+/%* etc. should be separated into separate lines. This will improve the quality of data presentation.

[238] Were there any differences between reference sequences and the sequences obtained from Sanger sequencing? In some sequences the similarity was not 100%, did the authors find some specific single nucleotide polymorphisms in 18s rRNA sequence? In addition, have you compared the sequences obtained from feeding ticks and questing ticks in order to find molecular differences between Trypanosoma spp. present in these two types of ticks? The comparative analysis of Trypanosoma 18s rRNA sequences obtained from feeding and questing ticks should be indicated in order to demonstrate what are the molecular differences or whether are any molecular differences between these sequences. Please add this information to the article.

[265-266] The sequences are yet not available in GenBank.

Discussion:

[328] Unnecessary gap in the discussion section.

Author Response

Response to Reviewer 2 Comments

The authors appreciate the positive feedback provided by reviewer 2 and their support for the publication of this manuscript.

Materials and Methods:

[171] Please indicate in which statistical program the statistical analyses were performed.

This has since been addressed in line 276-277.  

Results:

[189] There's no need to repeat the information about permit number as it was mentioned in Material and Methods section.

Thank you for bringing to our attention this duplicate. The permit number has since been removed from this sentence.

[201] The Table 1 can be divided in two separate tables - one for questing ticks, the other one for feeding ticks.

We thank the reviewer for their suggestion. However, as this was not a unanimous suggestion by all reviewers, we have retained the current format.

[202] The abbreviations L, N, A are not necessary - these words are not too long to not be placed in the table. I suppose it will be easier to read it.

Thank you for this suggestion. We have since removed these abbreviations under the ‘life stage frequency columns’ and replaced them with their full names.

[203] In my opinion the adults column should be above male and female columns as the division between sexes is made only in this stage of ticks.

Please observe the revised table which has separated the data for adult ticks into distinctive male and female columns.

[203] Do the authors have an information about the presence of questing ticks on specific sites? I.e. the larvae was collected from the leaves, whereas adult forms from clothes? In my opinion the information about the exact localization of questing ticks at specific life stage should be indicated like the host species in case of feeding ticks.

Questing ticks were opportunistically collected directly from our clothes and/or equipment (i.e. bags) onto which they had crawled. During the occasions in which we noticed ticks crawling towards us in the field, we collected them. We found no disparity between tick life stages here and therefore do not provide comment of this in the manuscript. All developmental stages had crawled onto clothing. Given their larger size and hence the ease at which nymphs and adults were detectable on the ground, these life stages were collected as and when observed. Please refer to lines 156-157.

[204] The information L n/qPCR+/%* etc. should be separated into separate lines. This will improve the quality of data presentation.

Table revised.

[238] Were there any differences between reference sequences and the sequences obtained from Sanger sequencing? In some sequences the similarity was not 100%, did the authors find some specific single nucleotide polymorphisms in 18s rRNA sequence? In addition, have you compared the sequences obtained from feeding ticks and questing ticks in order to find molecular differences between Trypanosoma spp. present in these two types of ticks? The comparative analysis of Trypanosoma 18s rRNA sequences obtained from feeding and questing ticks should be indicated in order to demonstrate what are the molecular differences or whether are any molecular differences between these sequences. Please add this information to the article.

We thank the reviewer for this suggestion. We have added the appropriate data to the results section, lines 515-526.

[265-266] The sequences are yet not available in GenBank.

As per lines 531-533, sequences have been deposited in GenBank (accession numbers MW881302-MW881349). They will be automatically released by GenBank upon publication of this manuscript.

Discussion:

[328] Unnecessary gap in the discussion section.

Thank you for pointing out this formatting error. It has since been revised.

Reviewer 3 Report

The manuscript by Krige et al., represents a survey of trypanosome DNA presence in questing and feeding Australian ticks. The authors used High-Resolution Melt qPCR for the detection and sequencing of amplicons for the confirmation that these are indeed trypanosomes and identification of species. Trypanosomes have been detected in both groups (feeding and questing) and at all studied developmental stages of ticks (larvae, nymphs, and adults), and the authors concluded that these belonged to 5 previously documented species endemic to Australia. From the presented data it appears quite plausible, that ticks are the vectors of these parasites and the authors satisfactorily discuss most observed details.

Even before reading the main text of the manuscript I was convinced that the data are worth publishing, however there are several issues requiring revision of the manuscript. The first issue is the wordiness, which significantly complicates reading. There are unnecessary words, sentences and even paragraphs distracting the reader. The abstract even from the formal point of view is much longer than it should be (I counted 288 words, whereas 200 is the maximum).

The second issue is incompleteness of the analysis and information. The material included samples from different tick species, but the statistics are summarized over them (except the number of analyzed specimens). The distribution of trypanosome species over the species of ticks and mammals (for feeding specimens) is also unclear. The authors identified species by BLAST, which would be OK in the case of 100% similarity, however there are as low as 95% identity levels. It would be more informative and illustrative to build a tree as on the fig 3. in https://doi.org/10.1016/j.ijppaw.2020.11.003 (reference 25 from the manuscript). Although the analyzed fragment is quite short, it still appears suitable for such an analysis. Moreover, the tree would be a great opportunity to show with what tick species these sequences were associated.

In addition, it is absolutely necessary to present a supplementary table, describing every positive sample (at least: tick species, developmental stage, status (feeding/questing), accession number(s), best blast hit, and identity percentage). Out of 71 qPCR-positive samples only 58 have been confirmed to contain trypanosome DNA, whereas it is not immediately explained what are the rest 13. Moreover, out of these 71 only 48 sequences have been submitted to GenBank, as judged by the range of the listed accession numbers. This is definitely unacceptable. All analyzed data must be made available (by the way, I did not find the data availability statement). At some point the authors mention that some of the sequences belonged not to Trypanosoma spp. but to "Bodo". This requires a more detailed description (needless to say, that it should be in the abovementioned supplementary table and the sequences submitted to GenBank). As judged by their previous paper (https://doi.org/10.1016/j.ttbdis.2020.101596, reference 22), the authors use the name "Bodo" for organisms belonging to three different orders of Kinetoplastea, i.e. as different from each other as from trypanosomatids. I insist on discussing this. Firstly, in the Material and Methods, where currently we see that qPCR is trypabosomatid-specific, which is definitely incorrect. It must be explained, that the primers work also on other Kinetoplastea, as has been shown before and this is exactly the reason, why confirmation by sequencing is mandatory. In the Results it must be mentioned, what were these 13 sequences. In the Discussion I suggest to provide an interpretation of the observed results and this is not what the authors have written before in the previous paper. Ticks consume water (https://doi.org/10.1038/s41598-020-63004-9 ) and this is quite a likely route for obtaining free-living kinetoplastids (or their DNA).

The third issue is the language, which should be improved. There are repeated words in a single sequences, awkward sentences etc.

Below are specific editing suggestions with indication of lines.

22: "invertebrate vectors" – just "vectors", since vertebrates cannot be their vectors

28-29: "All questing ticks…" the sentence does not belong to the abstract

36-41: these sentences are fully redundant

By the way, these three sentences are 81words (now 88 are exceeding the limit).

46: "class Kinetoplastida " > "class Kinetoplastea"

46: "that are of veterinary and medical significance" out of ~ 500 described species in the genus only a tiny proportion is of such significance, so revise as " and some of them are of veterinary and medical significance".

50:

51: "haematophagous species within the order Diptera" > haematophagous representatives (or members) of the order Diptera. Here it needs to be specified, whether this is about all trypanosomes or just the terrestrial ones. Otherwise, leeches (Hirudinea) must be mentioned as vectors for the trypanosomes of fish and frogs and sandflies (Psychodidae: Phlebotominae) for frogs. In addition to what is listed for terrestrial trypanosomes, the authors should include fleas (order Siphonaptera) as vectors of the subgenus Herpetosoma and blackflies (fam. Simuliidae) for some avian trypanosomes.

52-53: "and species within the family Reduviidae (triatomines)" this is a family of the order Hemiptera (suborder Heteroptera, true bugs), not Diptera! "species within the family" is an awkward wording, revise

62-63: repeated "species" in the sentence. Revise as " affirming it as critically endangered"

63-64: repeated "their"; the first one can be replaced with "this"

72: replace semicolon with comma

71-73: This is an incorrect description. The stages are the egg, larva, nymph and adult. 2-4 feed on blood, metamorphosis occurs between 2-3 and 3-4. 1-2 is hatching, 4-1 is oviposition.Revise accordingly.

89: "close genetic proximity" – this is an exaggeration. T. cruzi together with T. marinkellei belongs to the subgenus Shizotrypanum , the sister subgenus is Aneza (T. rangeli, T.conorhini, etc.) and T. noyesi is most closely related to the clade of T. wauwau/T.janseni/T.madeirae, with which it forms a clade sister to that of the two above subgenera. I would write "one of the relatives" (not much close, by the way).

93: " several different morphologies" morphology is an uncountable noun, here should be "morphotypes" (trypo-, epi-, and amastigotes)

120: double "within", revise

122: "ambient temperatures" > " ambient temperature"

127: "post" > after

130-131: "Questing ticks were bisected as previously described [22]. Feeding ticks were similarly processed." > The ticks were bisected as previously described [22].

133-138: Be concise. "…following manufacturers … DNA yield." > …with overnight digestion at 56°C and double elution with 30-40 μL of AE buffer.
138-140: "Extraction reagent blank controls were incorporated with each DNA extraction batch to demonstrate the absence of contamination during downstream amplification." > Each DNA extraction batch contained blank controls subsequently checked by PCR for contamination.

143-144: "For feeding ticks, individual adult, nymph and larvae (n = 72) ticks and four feeding larvae pools…" > For feeding ticks, individual adults, nymphs and larvae (n = 72) and four larvae pools…

146-147: just delete redundant "with amplification parameters followed in accordance with the cycling protocol"

151: Explain what is "no template control". What is the difference with PCR-grade H2O and "extraction reagent blanks"?

152: "Leishmania macropodum sp. nov." > Leishmania macropodum
It is not a new species anymore. "Sp. nov." can be added only in the publication with the original description of a species.

152-153: delete the sentence "Extraction reagent blanks were similarly assessed." And putthese blanks to the list in the previous sentence.

153-154: " Post amplification melt curve …" This is a redundant sentence. Species identification here has been done based on sequences.

157: Delete " and viewed using an LED light transilluminator." This is obvious.

158: Delete the sentence. It is also obvious.

159: "Positive samples" > Amplicons

160: Delete "as per the manufacturer’s instructions"

161-166: Delete the sentences. Nobody cares.

166-169: revise as: "Sequence identification was performed using BLAST against NCBI nr nucleotide database."

172 and 174: "infection" > prevalence (or infection rate)

175-176: incorrect formulation; should be "Significance level (alpha) was set to 0.05".

179: "ixodid ticks" > ticks (otherwise it is a tautology)

179: "were collected" > were collected in the UWR

180: "Trypanosoma" > Trypanosoma spp.

182: delete "within the UWR" (see above)

184: " spp.." > spp.

188: delete ", similarly located within the UWR," (see above)

192: add comma before "respectively"

194: "sampling … were" > sampling … was

Table 1: 1) spell out the life stages in the titles (then there is no need to explain them below);
2) separate the column "adults" into males and females, or present them as total, males, and females; 3) "life stage frequency" > life stage; 3) for every cell in the table present the following: positive/total (prevalence,%), then subtotals do not need their own subtitles (like" N n/qPCR+/%")

213: " based on the presence of" > by

214: "within" > with

215-216: "combined with the presence of a corresponding gel band of the correct amplicon size for Trypanosoma DNA." > "and the presence of an amplicon of ~ 250 bp"

216-217:" PCR positivity was 216 slightly higher…" - According to chi-squared test this difference is not significant (p = 0.5664)

228-229: " previously documented trypanosome species located within the region" > trypanosome species previously documented in the region

230: delete "assignment to the previously proposed"

231:" delete also the reference [15]

232: delete "Australian"

233-260: revise and shorten after presenting the data in a more illustrative form

261-262: "a mixed infection for T. copemani and T. vegrandis DNA." Either "a mixed infection with T. copemani and T. vegrandis" or "the presence of T. copemani and T. vegrandis DNAs"

265-266: submit missing sequences and add accession numbers

Table 2: transpose it to have four lines and two columns; it will fit better

269-271: revise both sentences; "DNA was highest" and "were highest …for DNA"make no sense.

280: delete "collected from vegetation"

281: delete " removed from marsupial hosts"

285: "Trypanosoma" > Trypanosoma spp.

285: "T. copmeni" > T.copemani

286-287: "from two different genera of ticks, within both questing and feeding ticks collected from south-west Australia" > from questing and feeding ticks (genera Amblyomma and Icoxodes) collected in south-west Australia

292: replace semicolon with comma

294-296: "to the Australian Trypanosoma life cycle, which is currently incomplete since no vector(s) have been established. " > to the Australian trypanosomes' life cycles, since the vectors have not been identified

302: delete "cosmopolitans;" Actually, they are not cosmopolitans, but endemics of a single, not that big continent

305-307: delete this sentence, repeating the information from the first paragraph of the Discussion

308-311: It is not that clear what the authors want to say. Many trypanosomes have much wider distribution, exceeding the limits of one continent. If the authors believe that the distribution of any Australian trypanosome species is wide, it should be described in detail. In addition, Why "this parasite", does it concern a particular species?

310: "widespread nature" does not read well, can be widespread distribution

312: " Trypanosoma DNA from feeding ticks was detected in all examined stages " > "Trypanosoma DNA was detected in all examined stages of both feeding and questing ticks"

315-316: delete " This similarly applies to questing ticks and hence"

317: these are definitely not "populations", can be categories or similar

317-320: revise the sentence as follows: Our observations are in agreement with the known biology of ticks: adults are the main life stage of ticks found on hosts, whilst immature nymphs are more frequent on vegetation.

322: replace colon with comma

325: "ixodid ticks" > ticks

325: "are the opportunistic host" > are opportunistic hosts

329-332: revise the sentence. "Initial positivity" is an incorrect formulation, also see above about this issue in general.

333: "high detection" is a nonsense

355-356: " Yet, co-infection with T. copemani and a shared sequence similarity for T. vegrandis and T. gilletti was detected" > Yet, co-infection with T. copemani and T. vegrandis or T. gilletti, (undistinguishable from each other by the amplified fragment) was detected

357: ", suggests that" > This suggests that

362-369: the whole paragraph is redundant. No need to discuss this here, especially given the very short sequences.

370-372: revise the sentence to make it grammatical

376: "were positive for Trypanosoma DNA, only one of these ticks was DNA positive " > contained Trypanosoma DNA, only one was positive for

377: " a noteworthy difference" – it is not, since this difference is not statistically significant

378-389, 393-396, 398-407 redundant text to be deleted. Instead of the last passage, write just one sentence explaining what is needed to verify this hypothesis.

397: delete "suggests"

Round 2

Reviewer 1 Report

Revised version is acceptable

Author Response

We thank the reviewer for their acceptance of our revised manuscript.

Reviewer 3 Report

The authors improved the manuscript by adding the supplementary data and revising the text, which now reads significantly better. However some changes are still pending (see below).

First of all, the Figure 1 has not been included into the manuscript, there is only the legend. I hope that this tree will be included and that it contains not only the sequences obtained in this work, but also reference sequences for all species under question from Genbank. Of note, for better resolution the full length of reference sequences can be preserved.

Concerning this part of the analysis, since I was significantly confused by the identity percentage for the sequences, I extracted the region amplified in T. copemani with the primers TrypF and TrypR (from the cited paper) and found out that it is relatively conservative and the overwhelming majority of trypanosome species sequenced to date fall within 95% identity range (except some aquatic ones). This demonstrates that the methodology used here may be inadequate. As judged by authors' comments, there could be an unjustifiably high error rate in the obtained sequences (they claim 5%). The papers about bacteria cited in the comments are absolutely irrelevant. We discuss here the 18S rRNA gene, which is highly conservative within trypanosomes and 95% identity threshold, as I mentioned above allows identification only up to genus. I have downloaded the sequences of Australian trypanosomes and the INTERspecific difference between the most distant by this region does not exceed 2% (the most divergent is Trypanosoma sp. ANU2).

The authors should explain well what really happened, because as by now it does not seem convincing. We do not even know how many species were there (who said that the whole diversity of Australian trypanosomes has been already described)? I cannot exclude that some of the errors actually represent ambiguous bases thereby decreasing identity levels. However, such cases could also mean mixed samples. Neither me nor any other readers should guess. The information must be presented as clearly as possible. Now this represents the most serious concern.

The Suppl. Fig. 1 shows a tree with a large cluster of Bodo spp. (order Eubodonida) rooted with a sequence, which is most closely related to that of Parabodo caudatus (order Parabodonida) and definitely belongs to the genus Parabodo, not Bodo, regardless of what is stated in the record. Since this is not a sequence belonging to the dataset, its presence there is misleading. Instead it would be better to use a properly identified sequence of Parabodo caudatus or related species.

Now concerning the text.

>>22: "invertebrate vectors" – just "vectors", since vertebrates cannot be their vectors
> We have retained ‘invertebrate vectors’ as vertebrates, such as bats, have been demonstrated as vectors for trypanosomes.

As far as I know, vampire bats do not live in Australia. If the authors insist on writing "invertebrate vectors", then they must explain, why they exclude the vertebrate ones. In principle, the discussed sentence is not about vectors in general, but about those for Australian trypanosomes. I do not understand why to complicate the things.

>>46: "class Kinetoplastida " > "class Kinetoplastea"
>A frequent oversight (https://doi.org/10.1186/s13071-017-2204-7; https://doi.org/10.1038/srep35826; https://doi.org/10.1080/15476286.2018.1564463). Our sincere appreciation for bringing this to our attention and it has since been amended, line 39.

It has been amended in a wrong way. Instead of correcting the name of the cllass, the authors changed the taxonomic rank. Order Kinetoplastida corresponds to an obsolete version of the classification. Since 2004 there is class Kinetoplastea, containing 5 orders, of which one is Trypanosomatida. See the following papers:
10.1099/ijs.0.63081-0
10.1111/jeu.12691
10.1098/rsob.20040

>>52-53: "and species within the family Reduviidae (triatomines)" this is a family of the order Hemiptera (suborder Heteroptera, true bugs), not Diptera! "species within the family" is an awkward wording, revise

>Revised.

Now the text reads as Siphonaptera and bugs are part of Diptera. Semicolons do not help much. Mentioning of Hemiptera/Heteroptera did not appear.

>>71-73: This is an incorrect description. The stages are the egg, larva, nymph and adult. 2-4 feed on blood, metamorphosis occurs between 2-3 and 3-4. 1-2 is hatching, 4-1 is oviposition. Revise accordingly.

>Our original description is correct.

No, it is not. You wrote: "four developmental stages, three of which require a blood meal from a vertebrate host prior to metamorphosing to the next life stage". Well, larva metamorphoses to nymph, nymph metamorphoses to adult, but adult does not metamorphose to egg! I suggest to revise the text with using a more general word like "switching" instead of "metamorphosing".

For the Table 1, the following suggestion with detailing the data has been ignored:
>> 3) for every cell in the table present the following: positive/total (prevalence,%), then subtotals do not need their own subtitles (like" N n/qPCR+/%")

>>294-296: "to the Australian Trypanosoma life cycle, which is currently incomplete since no vector(s) have been established. " > to the Australian trypanosomes' life cycles, since the vectors have not been identified

>Revised. Line 640.

Revised insufficiently. There are several species of trypanosomes. Formally, each of them has its own life cycle, even if one cannot find any differences at the moment. So, there should be "life cycles", not "life cycle".

Comments to the new text (some were overlooked in the previous version):

36-37: " This work has confirmed Australian ticks are capable" add "that" after "confirmed" for easier reading

42: "could act as potential vectors" – this is a vague formulation with double uncertainty. Revise as, e.g. "likely represent vectors".

177-178: "Phylogenetic analyses of the amplified section of 18S rRNA was conducted for ticks that contained Trypanosoma" This is incorrect, the analysis has been made for protists, not for ticks.

256 and elsewhere: "between 95.24-100%" the dash already indicates the range, the word "between" is redundant

278-279: "ticks shared between 98.46-100% similarity to Trypanosoma sp. ANU2" The authors apparently did not compare ticks with trypanosomes, but trypanosomes to trypanosomes.

279-280: "One questing tick displayed a mixed infection 279 with T. copemani and T. vegrandis DNA." "infection ... with DNA" is nonsense. Infecting agents are trypanosomes, not DNAs. So should be "presence of T. copemani and T. vegrandis DNA(s)" or similar.

286: "T. noyesi positive ticks" > T. noyesi-positive ticks

286-287: "were of the species A. triguttatum" > belonged to the species A. triguttatum,

291-292: 'no significant nucleotide diversity was correlated with different tick species."  - Not exactly clear what is meant here. Is it about distribution of trypanosome species over those of ticks? Then there should be some illustration demonstrating this, otherwise this is a mere assertion.

294: "undescribed soil eukaryotes" – it is obvious that such sequences are closely related to (genuine) Bodo spp. and therefore are also Bodo spp. (of note, there are no other genera in the order Eubodonida). So just omit these three words

297-298: "Amplified sequences generated from this investigation that were unique for the present study" > New sequences obtained in the present study

364: "prevelant" > prevalent

377: "reviewed" > reported (or documented)

378-379: " Whilst further evaluation concerning the occurrence of Bodo DNA within ticks was beyond the scope of this study," I agree that this was beyond of the scope, but there is no need to state that. Please, delete this passage to make the sentence more concise. Also, please, add a relevant reference.

382: "this free-living protozoan is" > these free-living protozoa are

385: delete "in infection rates", now it is redundant given the above text.

387-388: "presence for Trypanosoma DNA" > presence of Trypanosoma DNA

Author Response

The authors are pleased that reviewer 1 and reviewer 2 are satisfied with the previous revisions and welcome their support for the publication of our manuscript.

Please find our responses to reviewer 3’s comments below.

The authors improved the manuscript by adding the supplementary data and revising the text, which now reads significantly better. However some changes are still pending (see below).

First of all, the Figure 1 has not been included into the manuscript, there is only the legend. I hope that this tree will be included and that it contains not only the sequences obtained in this work, but also reference sequences for all species under question from Genbank. Of note, for better resolution the full length of reference sequences can be preserved.

The file for Figure 1 has been uploaded as per the journal’s instructions. We have since added the figure to the revised manuscript as it appears the inclusion of this figure may have been overlooked by the journal editing team.

Concerning this part of the analysis, since I was significantly confused by the identity percentage for the sequences, I extracted the region amplified in T. copemani with the primers TrypF and TrypR (from the cited paper) and found out that it is relatively conservative and the overwhelming majority of trypanosome species sequenced to date fall within 95% identity range (except some aquatic ones). This demonstrates that the methodology used here may be inadequate. As judged by authors' comments, there could be an unjustifiably high error rate in the obtained sequences (they claim 5%). The papers about bacteria cited in the comments are absolutely irrelevant. We discuss here the 18S rRNA gene, which is highly conservative within trypanosomes and 95% identity threshold, as I mentioned above allows identification only up to genus. I have downloaded the sequences of Australian trypanosomes and the INTERspecific difference between the most distant by this region does not exceed 2% (the most divergent is Trypanosoma sp. ANU2).

We feel the methodology used in this paper is justified, as we have not relied solely upon sequencing. As mentioned, we also utilised melt curve analyses (can differentiate between species of trypanosome) as demonstrated by Keatley et al., 2020 and Krige et al., 2021. Consolidating the information obtained from melt curve analysis with sequencing data confirms species identity.  

The authors should explain well what really happened, because as by now it does not seem convincing. We do not even know how many species were there (who said that the whole diversity of Australian trypanosomes has been already described)? I cannot exclude that some of the errors actually represent ambiguous bases thereby decreasing identity levels. However, such cases could also mean mixed samples. Neither me nor any other readers should guess. The information must be presented as clearly as possible. Now this represents the most serious concern.

The methodology is explained in detail, and at no stage have we suggested that the whole diversity of Australian trypanosomes has been described. We have indicated the presence of a mixed sample and have since included a sentence to confirm that the presence of additional mixed infections cannot be ruled out. Line 372-384. At no point in this manuscript have we presented anything other than the data generated in this study.

The Suppl. Fig. 1 shows a tree with a large cluster of Bodo spp. (order Eubodonida) rooted with a sequence, which is most closely related to that of Parabodo caudatus (order Parabodonida) and definitely belongs to the genus Parabodo, not Bodo, regardless of what is stated in the record. Since this is not a sequence belonging to the dataset, its presence there is misleading. Instead it would be better to use a properly identified sequence of Parabodo caudatus or related species.

We thank reviewer 3 for their suggestion. The supplemental figure has been rooted with the Bodo sp. detected from within the same region sampled (https://doi.org/10.1186/s13071-019-3370-6). We do not understand why the use of this sequence presents an issue. Nevertheless, we have since exchanged this root sequence for Parabodo caudatus in keeping with reviewer 3’s instruction.

Now concerning the text.

>>22: "invertebrate vectors" – just "vectors", since vertebrates cannot be their vectors
> We have retained ‘invertebrate vectors’ as vertebrates, such as bats, have been demonstrated as vectors for trypanosomes.

As far as I know, vampire bats do not live in Australia. If the authors insist on writing "invertebrate vectors", then they must explain, why they exclude the vertebrate ones. In principle, the discussed sentence is not about vectors in general, but about those for Australian trypanosomes. I do not understand why to complicate the things.

As requested, we have removed ‘invertebrate vectors’ in place of ‘vectors’. Line 21.

>>46: "class Kinetoplastida " > "class Kinetoplastea"
>A frequent oversight (https://doi.org/10.1186/s13071-017-2204-7; https://doi.org/10.1038/srep35826; https://doi.org/10.1080/15476286.2018.1564463). Our sincere appreciation for bringing this to our attention and it has since been amended, line 39.

It has been amended in a wrong way. Instead of correcting the name of the cllass, the authors changed the taxonomic rank. Order Kinetoplastida corresponds to an obsolete version of the classification. Since 2004 there is class Kinetoplastea, containing 5 orders, of which one is Trypanosomatida. See the following papers:
10.1099/ijs.0.63081-0
10.1111/jeu.12691
10.1098/rsob.20040

Thank you for the further correction. We have amended the sentence to ‘class Kinetoplastea’. Line 39.

>>52-53: "and species within the family Reduviidae (triatomines)" this is a family of the order Hemiptera (suborder Heteroptera, true bugs), not Diptera! "species within the family" is an awkward wording, revise

>Revised.

Now the text reads as Siphonaptera and bugs are part of Diptera. Semicolons do not help much. Mentioning of Hemiptera/Heteroptera did not appear.

Agreed. We have revised lines 46-48.

>>71-73: This is an incorrect description. The stages are the egg, larva, nymph and adult. 2-4 feed on blood, metamorphosis occurs between 2-3 and 3-4. 1-2 is hatching, 4-1 is oviposition. Revise accordingly.

>Our original description is correct.

No, it is not. You wrote: "four developmental stages, three of which require a blood meal from a vertebrate host prior to metamorphosing to the next life stage". Well, larva metamorphoses to nymph, nymph metamorphoses to adult, but adult does not metamorphose to egg! I suggest to revise the text with using a more general word like "switching" instead of "metamorphosing".

Good point! We have now revised line 69 as recommended.

For the Table 1, the following suggestion with detailing the data has been ignored:
>> 3) for every cell in the table present the following: positive/total (prevalence,%), then subtotals do not need their own subtitles (like" N n/qPCR+/%")

Table 1 was originally modified taking into consideration comments from all of the reviewers. As this suggested change was not unanimous or consistent with other requested changes, we have maintained the current format of Table 1.

>>294-296: "to the Australian Trypanosoma life cycle, which is currently incomplete since no vector(s) have been established. " > to the Australian trypanosomes' life cycles, since the vectors have not been identified

>Revised. Line 640.

Revised insufficiently. There are several species of trypanosomes. Formally, each of them has its own life cycle, even if one cannot find any differences at the moment. So, there should be "life cycles", not "life cycle".

Line 298 of the manuscript stated life cycle(s). Trypanosoma implies the species within, and the presence of life cycle(s) is noted. 

Comments to the new text (some were overlooked in the previous version):

36-37: " This work has confirmed Australian ticks are capable" add "that" after "confirmed" for easier reading

Changed - Line 32.

42: "could act as potential vectors" – this is a vague formulation with double uncertainty. Revise as, e.g. "likely represent vectors".

Changed - Line 33-34.

177-178: "Phylogenetic analyses of the amplified section of 18S rRNA was conducted for ticks that contained Trypanosoma" This is incorrect, the analysis has been made for protists, not for ticks.

Changed - Line 172-173.

256 and elsewhere: "between 95.24-100%" the dash already indicates the range, the word "between" is redundant

Changed - Lines 238, 241, 244, 249.

278-279: "ticks shared between 98.46-100% similarity to Trypanosoma sp. ANU2" The authors apparently did not compare ticks with trypanosomes, but trypanosomes to trypanosomes.

Changed - Line 249-250.

279-280: "One questing tick displayed a mixed infection 279 with T. copemani and T. vegrandis DNA." "infection ... with DNA" is nonsense. Infecting agents are trypanosomes, not DNAs. So should be "presence of T. copemani and T. vegrandis DNA(s)" or similar.

Changed - Line 250-251.

286: "T. noyesi positive ticks" > T. noyesi-positive ticks

Changed - Lines 256, 258, 260.

286-287: "were of the species A. triguttatum" > belonged to the species A. triguttatum,

Changed - Line 412.

291-292: 'no significant nucleotide diversity was correlated with different tick species."  - Not exactly clear what is meant here. Is it about distribution of trypanosome species over those of ticks? Then there should be some illustration demonstrating this, otherwise this is a mere assertion.

Sentence added and is in keeping with reviewer 2’s suggestions.  

294: "undescribed soil eukaryotes" – it is obvious that such sequences are closely related to (genuine) Bodo spp. and therefore are also Bodo spp. (of note, there are no other genera in the order Eubodonida). So just omit these three words

Changed - Line 264.

297-298: "Amplified sequences generated from this investigation that were unique for the present study" > New sequences obtained in the present study

Changed - Line 266.

364: "prevelant" > prevalent

Changed - Line 333.

377: "reviewed" > reported (or documented)

Changed - Line 344.

378-379: " Whilst further evaluation concerning the occurrence of Bodo DNA within ticks was beyond the scope of this study," I agree that this was beyond of the scope, but there is no need to state that. Please, delete this passage to make the sentence more concise. Also, please, add a relevant reference.

This sentence was included specifically for reviewer 3, who had proposed further information concerning Bodo. With this sentence in place, we have informed the reader that there is no further evaluation of Bodo in this study.

382: "this free-living protozoan is" > these free-living protozoa are

Changed - Line 351.

385: delete "in infection rates", now it is redundant given the above text.

Changed - Line 354.

387-388: "presence for Trypanosoma DNA" > presence of Trypanosoma DNA

Changed - Line 356.